# Newborn Screening for Metachromatic Leukodystrophy: A Systematic Literature Review

**DOI:** 10.3390/ijns11040103

**Published:** 2025-11-05

**Authors:** Lucia Laugwitz, Andrew Shenker, Erica F. Sluys, Stéphane Pintat, David Whiteman, Charlotte Chanson

**Affiliations:** 1Neuropaediatrics, General Paediatrics, Diabetology, Endocrinology and Social Paediatrics, University Hospital Tübingen, University of Tübingen, 72076 Tübingen, Germany; 2Independent Clinical Consultant, Pennington, NJ 08534, USA; andrewshenker@comcast.net; 3And You SARL, 1260 Nyon, Switzerland; erica@andyoueducation.com; 4Oxford PharmaGenesis, Oxford OX13 5QJ, UK; stephane.pintat@pharmagenesis.com; 5Independent Clinical Consultant, Cape Elizabeth, ME 04107, USA; dahw.mobile@gmail.com; 6Orchard Therapeutics, London W6 8PW, UK

**Keywords:** newborn screening, metachromatic leukodystrophy, systematic literature review

## Abstract

A systematic literature review was conducted to evaluate the emerging evidence on newborn screening (NBS) for metachromatic leukodystrophy (MLD; MIM #250100). The review focuses on (1) screening assay performance, (2) diagnostic confirmation methods and care pathways, (3) feasibility of population-based identification, and (4) the impact of early diagnosis and treatment on health outcomes. Electronic databases were searched in February 2025, and supplementary searches were performed up to 17 June 2025, for articles referencing NBS for MLD and treatments for MLD; 52 publications were eligible for inclusion. Nationwide NBS for MLD is currently carried out in Norway and large prospective pilots are running in Germany, Austria, Italy and the US. MLD meets established Wilson and Jungner criteria, with a reliable screening algorithm, established confirmatory diagnostics, and actionable care pathways. There is ongoing work to develop tools to predict disease severity and subtype. Early intervention—via gene therapy for early-onset MLD and hematopoietic stem cell transplantation (HSCT) for late-onset forms—significantly improves outcomes when initiated before symptom onset. This review provides the first comprehensive synthesis of the evidence supporting MLD for inclusion in NBS programs, underscoring the public health value of early identification and intervention.

## 1. Introduction

Metachromatic leukodystrophy (MLD; MIM #250100) is a devastating lysosomal storage disorder (LSD) characterized by progressive demyelination and neurodegeneration. It is caused by variants in the *ARSA* gene (MIM *607574), which encodes the enzyme arylsulfatase A (ARSA). The deficient enzyme activity results in accumulation of sulfatides throughout the body, particularly in the nervous system [1,2]. The buildup exerts a toxic effect on myelin-producing cells and neurons, manifesting clinically as rapid motor and cognitive decline, followed by premature death, particularly in its early-onset forms. MLD is a rare disease with a reported birth prevalence of approximately 1 per 100,000 live births (range: 1 per 40,000 to 160,000) [3,4].

Based on age at symptom onset, MLD is classified into four subtypes: late-infantile [LI], early-juvenile (EJ), late-juvenile (LJ), and adult (A) [5]. These subtypes broadly fall into two phenotypic categories: early-onset (LI and EJ) and late-onset (LJ and A), with early-onset forms comprising the vast majority of cases [1,6,7]. Patients with LI-MLD, the most common subtype, typically experience symptom onset before 30 months of age; EJ-MLD onset appears between 30 months and 7 years; LJ-MLD onset appears between 7 and 16 years of age; and adult MLD onset appears over 16 years of age.

Previously reported genotype–phenotype correlations support this classification and are closely linked to residual ARSA enzyme activity. Patients with LI-MLD almost invariably harbor biallelic protein-truncating variants (PTVs) in the *ARSA* gene that result in minimal residual enzyme activity (typically <1% of normal). Patients with EJ-MLD often carry one PTV and one missense variant, or two missense variants in *trans*, typically yielding higher residual activity [8,9,10]. Individuals with late-onset MLD subtypes (LJ or A) frequently harbor biallelic missense variants, a few of which are exclusively associated with late-onset phenotypes, and produce comparably higher levels of residual ARSA activity [9,11].

Currently, most children with MLD are diagnosed only after they exhibit overt clinical symptoms, by which point the window for effective intervention with disease-modifying therapies has closed. The earliest signs of MLD—such as stagnation in acquiring developmental milestones and changes in gait or behavior—are often subtle and nonspecific, making clinical recognition challenging [12,13,14]. As a result, diagnosis is frequently delayed for months or even years, during which time patients experience irreversible neurological damage. With very few exceptions the only patients with early-onset MLD identified in the pre-symptomatic stage are those with an older affected sibling [15,16], highlighting the critical need for proactive screening approaches. Autologous hematopoietic stem cell lentiviral gene therapy (atidarsagene autotemcel; arsa-cel) is a novel disease-modifying therapy that is now considered standard of care for patients with pre-symptomatic LI-MLD and pre-symptomatic or early-symptomatic EJ-MLD [11,15,17,18,19]. Importantly, the most favorable motor and cognitive outcomes with arsa-cel are achieved when treatment is administered during the pre-symptomatic stage [15]. Arsa-cel was approved in 2020 by the European Medical Agency, in 2021 by the UK Medicines and Healthcare Products Regulatory Agency, and in 2024 by the US Food and Drug Administration [20,21,22]. For patients with late-onset MLD, allogeneic hematopoietic stem cell therapy (HSCT) has been the standard of care for the past two decades, also showing the most significant benefits when administered before symptom onset [23,24,25,26,27,28].

Given the rapid progression and lethality of early-onset MLD—and the potential to drastically alter the natural disease course of most patients with MLD with pre-symptomatic treatment—this systematic literature review was conducted to evaluate the emerging evidence surrounding newborn screening (NBS) for MLD, focusing on (1) the performance of screening assays, (2) diagnostic confirmation methods and care pathways, (3) the ability to identify affected infants in a population-based screening program, and (4) the impact of early identification and treatment on health outcomes. Following the publication of a key pilot study demonstrating the feasibility of NBS for MLD [29], momentum has built in the field. Two prospective pilot studies from Germany and Italy have recently been published [30,31], and several more are currently underway or planned [32,33,34,35].

The Advisory Committee on Heritable Disorders in Newborns and Children (ACHDNC) Evidence Review Group has published an evidence-based review following a nomination to add MLD to the US Recommended Uniform Screening Panel (RUSP) [36], and three US states—Illinois, Minnesota and Pennsylvania—have approved the inclusion of MLD to their NBS panels [37,38,39]. In 2024, Norway became the first country to implement national NBS for MLD [40].

Conducting a literature review at this stage provides a valuable opportunity to synthesize the current evidence supporting MLD NBS, highlight critical knowledge gaps requiring further investigation, and offer timely, actionable insights to inform both the implementation of MLD NBS programs and the direction of future research.

## 2. Methods

### 2.1. Search Strategy

Publications were identified by searches of electronic databases, including MEDLINE (In-Process & Other Non-Indexed Citations and Ovid MEDLINE [1946–present]), Embase (1974–present), and the Cochrane Library (Cochrane Database of Systematic Reviews; Database of Abstracts of Reviews of Effects; Cochrane Central Register of Controlled Trials; Cochrane Methodology Register; NHS Economic Evaluation Database; Health Technology Assessment Database; American College of Physicians Journal Club). Two search strategies were developed: the first combined free text and Medical Subject Headings (MeSH) related to NBS with terms for MLD, and the second combined MLD terms with those related to treatments. Both strategies were designed to maximize sensitivity and ensure comprehensive coverage of relevant literature. The search was restricted to English-language literature published from 2015 onwards and was not limited by study design, country, or publication type. A detailed account of the search strings is provided in Appendix A. Electronic database searches were run on 19 February 2025.

In addition to the primary database searches, supplementary searches were conducted to identify gray literature and recent developments not captured in indexed databases. These included structured Google searches and hand-searching of abstract books and proceedings of key international congresses in the fields of lysosomal storage diseases on inborn errors of metabolism, held within the past 2 years, specifically the We’re Organizing Research on Lysosomal Diseases Symposium (WORLD*Symposium*), the Society for the Study of Inborn Errors of Metabolism (SSIEM) Annual Symposium, the International Symposium on MPS and Related Diseases (ISMPS), and the International Congress of Inborn Errors of Metabolism (ICIEM). Supplementary searches were performed between 9 April 2025 and 17 April 2025. Manual searches were also conducted on 17 June 2025 to capture any recent literature of interest, published after the cutoff date of the initial searches (19 February 2025).

The review followed the 2020 Preferred Reporting Items for Systematic Reviews and Meta-Analyses (PRISMA) guidelines [41], and the protocol was registered with the international Prospective Register of Systematic Reviews (PROSPERO; CRD420251021491).

Search results were downloaded into EndNote X9 (Clarivate Analytics, Philadelphia, PA, USA) and deduplicated before eligibility screening. Citations were transferred to Rayyan, a web-based tool designed to facilitate collaborative screening.

### 2.2. Publication Eligibility

The inclusion criteria, outlined in Appendix A, were designed to include studies that described or evaluated NBS or diagnostic approaches for MLD, and studies that reported the impact of early or pre-symptomatic treatment for MLD. Studies were eligible regardless of intervention, comparator, or study design, providing that they reported outcomes related to the performance (e.g., sensitivity, specificity, predictive values), feasibility, scalability, or cost-effectiveness of NBS assays, or considered the ethical, legal, or social aspects of NBS. Studies were considered eligible if they addressed at least one of the predefined inclusion criteria (e.g., screening assay performance, confirmatory diagnostics, feasibility, or clinical outcomes), even if not all criteria were reported. Data were also included on methods used to confirm diagnosis, including false positive/negative rates and methods used to identify and assess patients with MLD during population-based screening. Studies assessing the impact of early identification and treatment on clinical outcomes such as motor and cognitive functions, and treatment effectiveness were also eligible. Studies were also included if they reported the impact of NBS on time to diagnosis.

Titles and abstracts were screened independently by two reviewers against the predefined inclusion criteria (Appendix A). Discrepancies were resolved through discussion or adjudication by a third reviewer. Reasons for study selection were documented, and exclusions were based on the eligibility criteria. Full-text articles meeting the inclusion criteria were screened to confirm eligibility and shortlisted for data extraction. Reasons for exclusion at both the title/abstract and full-text stages were documented in a PRISMA-compliant format. Articles identified via supplementary searches were assessed using the same inclusion criteria.

### 2.3. Data Extraction

Data extraction was performed using a standardized template developed and piloted on a small subset of articles. Key data elements included study characteristics (e.g., author, year, country, design), details of the screening algorithms (e.g., method, performance metrics), confirmatory diagnosis details, the number and characteristics of screened and diagnosed infants, follow-up strategies, and health outcomes associated with early detection and treatment. Each study was extracted by a primary reviewer and checked for accuracy and completeness by a second reviewer.

### 2.4. Quality Assessment

The methodological quality and risk of bias of each study were assessed by an independent reviewer using validated tools appropriate to the study design. For randomized controlled trials, we used the Cochrane Risk of Bias 2 (RoB 2) tool; for non-randomized studies, we applied the Risk Of Bias In Non-randomized Studies of Interventions (ROBINS-I) tool; and for observational designs, we employed the Newcastle–Ottawa Scale (NOS). For case reports and expert opinion papers, we used the relevant checklists developed by the Joanna Briggs Institute. Quality assessment was conducted for full-text publications only; congress abstracts were excluded from assessment owing to limited information available to make a valid judgment. Quality ratings were used descriptively and did not determine study inclusion.

## 3. Results

The screening and selection process is summarized in the PRISMA flow diagram (Figure 1). In total, 454 records were identified from two systematic searches (Appendix A (NBS) and Appendix A (treatment)). Only conference abstracts with no corresponding full-text publication were included. After exclusions and additions following supplementary searches, a total of 52 publications (40 full-text articles [1,9,10,11,15,16,17,23,24,25,26,27,29,30,31,32,33,34,42,43,44,45,46,47,48,49,50,51,52,53,54,55,56,57,58,59,60,61,62,63] and 12 conference abstracts [64,65,66,67,68,69,70,71,72,73,74,75]) were eligible for data extraction. Of the full-text articles, 17 were assessed for methodological quality and risk of bias using tools appropriate to their study design. Quality assessment was not performed on congress abstracts and studies with designs outside the scope of the selected tools (e.g., economic models, feasibility studies, assay development studies, and Delphi panels). No randomized controlled trials were identified. Three observational studies were rated as good quality [23,24,50] and eight were rated as poor quality with the NOS [9,16,25,26,31,34,42,56]. Four non-randomized interventional studies were assessed using the ROBINS-I tool; three were judged as having a moderate risk of bias [15,30,47] and one had a serious risk of bias [48]. One case report [49] and one systematic review [27] were assessed as having a low risk of bias using the relevant JBI checklists.

### 3.1. Availability of Assays for MLD Diagnosis and Screening, NBS Algorithm Design

The diagnosis of MLD has traditionally involved measurement of ARSA activity in blood leukocytes and sulfatide levels in urine and sequencing of the *ARSA* gene [1,11]. Other than a single pilot study to measure ARSA protein in dried blood spot (DBS) samples from newborns using an immunoassay [68], early efforts to develop assays for MLD NBS have focused on high-throughput quantification of multiple sulfatide species and ARSA activity in DBS using liquid chromatography-tandem mass spectrometry (LC-MS/MS) [29,43]. Several sulfatide species, including C16:0, C16:0-OH, C16:1, C16:1-OH, C18:0 and C24:1, are elevated in the blood of individuals with MLD compared with healthy individuals [29,42,65,66] and methods for the rapid quantification of one or more sulfatides via LC-MS/MS for first-tier screening in MLD have been well established and optimized for use in NBS pilot programs [29,30,31,32,33,34,67]. The novel LC-MS/MS method that was developed and optimized to measure ARSA activity in leukocytes and DBS samples using a deuterated natural sulfatide substrate provides superior specificity, sensitivity and precision to traditional fluorometric assays utilizing an artificial substrate [43]. Moreover, it offers a high-throughput approach applicable for large scale NBS [64]. Measuring ARSA activity in DBS samples was made possible by optimizing a method for extraction and size-exclusion purification of the enzyme [43,62].

Genetic sequencing using DBS samples as a third tier can drastically reduce or even eliminate false positives during screening [44,76] and is conducted for DBS samples with low ARSA activity to identify individuals with MLD based on the presence of variants in the *ARSA* gene. This approach not only excludes samples from healthy carriers of a single *ARSA* variant from being considered as screen positives but also differentiates individuals with MLD from those with related disorders as described below [1,7,10].

As is common practice in NBS for other LSDs a multi-tiered algorithm is warranted for MLD for optimal sensitivity and specificity: low ARSA activity on its own is not necessarily indicative of MLD. Carriers of one disease-causing *ARSA* variant and individuals harboring pseudodeficiency alleles exhibit reduced ARSA activity but do not display clinical signs of MLD [54]. In addition, biochemical abnormalities in ARSA activity and/or sulfatides that mimic those in MLD are found in individuals with multiple sulfatase deficiency (MSD; MIM #272200) and saposin B deficiency (MIM #249900), ultrarare disorders due to biallelic variants in different genes, *SUMF1* (the gene encoding formylglycine-generating enzyme) and *PSAP* (the gene encoding prosaposin), respectively [1,8,73]. Both disorders are characterized by high sulfatides, but samples from patients with saposin B deficiency are not expected to show abnormally low ARSA activity in standard assays that contain detergent [43]. Given the lack of disease-modifying treatments for MSD or saposin B deficiency, individuals with those disorders are currently not candidates for inclusion in NBS protocols.

### 3.2. Feasibility of Multi-Tier NBS Programs for the Detection of MLD

Recent pilot studies have assessed the feasibility of NBS for the detection of MLD in infants from Austria, France, Germany, Italy, the UK, and the US, and all have used a multi-tiered algorithm (Table 1) [29,30,31,32,33,34,67,77].

#### 3.2.1. Retrospective Studies

Hong et al. 2021 [29] conducted a retrospective study of newborn DBS to assess the feasibility of detecting MLD with a two-tier biochemical screening, followed by genetic sequencing of the *ARSA* gene. Ultraperformance LC-MS/MS detection of C16:0 and subsequent measurement of ARSA activity was performed in de-identified DBS samples from 27,335 newborns [29]. The C16:0 cutoff value of ≥0.170 μM was established based on archived DBS samples from 15 patients with confirmed MLD (10 LI and five juvenile onset) and 2000 random newborns; to achieve 100% sensitivity (i.e., identification of all 15 patients with MLD). ARSA activity of <20% of the daily mean in similarly stored control samples was adopted as the cutoff for the second-tier assay, based on previous work [43]. First-tier screening identified 195 (0.71%) samples with C16:0 ≥0.170 μM. Of those, 122 DBS samples had been stored for ≤3 months at room temperature and therefore ARSA activity was determined. Two samples had an ARSA activity below the cutoff (<20% of daily mean) and normal activities of three other sulfatases known to be deficient in MSD, excluding the possibility of that disorder. Subsequent *ARSA* sequencing of DBS for these two samples identified a single pathogenic *ARSA* variant in one, interpreted as being from a carrier, and two pathogenic *ARSA* variants in the second sample, interpreted as possibly being from an infant with MLD. However, because the retrospective study design used de-identified DBS samples the diagnosis could not be confirmed [29]. This study concluded that NBS for MLD is feasible, highly sensitive, and specific for MLD. In addition, it quantified the benefit of each tier: a two-tier approach reduced the screen positive rate from the first-tier sulfatide assay alone (0.71%) to achieve an extremely low two-tier screen positive rate (0.0073%). Adding genetic sequencing of *ARSA* further reduced the screen positive rate (0.0037%).

A pre-pilot study in Manchester, UK assessed the feasibility of the same two-tier biochemical screening approach as reported by Hong et al. 2021 [29], including the assessment of reported cutoff values for C16:0 and ARSA (≥0.170 μM; alternatively expressed as ≥1.8 multiples of the median [MoM] and <20% mean of daily controls, respectively) in 3687 NBS samples [34]. During validation of the ARSA activity assay that was conducted prior to the pre-pilot study, 2 of 120 newborn DBS had enzyme activities below the initial cutoff value. To understand the reason for the reduced ARSA activity, measurement of C16:0 and *ARSA* gene sequencing was performed. One of the two samples was found to harbor a homozygous pathogenic variant known to be associated with LI-MLD. This sample also had the lowest C16:0 sulfatide result reported to date (0.150 μM), which was below the initial cutoff value, which had been adopted from the literature. Special approval from UK research and health authorities was urgently obtained to de-anonymize the DBS sample based on strong evidence of LI-MLD and potential eligibility for gene therapy.

Eleven of 3687 NBS samples analyzed in the pre-pilot study were first-tier screen positive with elevated sulfatides, but all were second-tier screen negative based on ARSA activity. However, two of the 20 positive control DBS samples from patients with confirmed MLD had C16:0 values below the initial cutoff value of 0.170 μM, underscoring the need to refine this value [34]. It was recommended that future pilot studies consider setting the Cl6:0 cutoff value to ≥0.150 μM (≥1.65 MoM) [34]. Preliminary data from this same study also suggested that C16:1-OH may be a better predictor for MLD than C16:0 alone [34].

As a consequence, the addition of C16:1-OH, as well as the analytical performance of the sulfatide screening assay as a first-tier assay was assessed in an analysis of pilot program data from several centers [32]. This study used cutoff values based on 40 DBS samples from patients with confirmed MLD; these values were expressed as MoM values to enable comparison across centers. Cutoff values for C16:0 and C16:1-OH were ≥1.65 MoM and ≥2.70 MoM, respectively. After screening 135,824 DBS samples of newborns, the combination of C16:0 and C16:1-OH was found to be superior to C16:0 or C16:1-OH alone as a first-tier biomarker for MLD, with a false positive rate estimated at 0.031%, compared with 1.8% for C16:0 alone and 0.048% for C16:1-OH alone.

#### 3.2.2. Prospective Studies

The first prospective NBS pilot utilizing a three-tiered approach began in Germany in 2021 [30,77]. Initial results from DBS samples collected from 109,259 newborns identified three screen-positive individuals who were subsequently confirmed to have MLD [30]. During the first-tier testing, elevated sulfatide levels (C16:0 ≥ 0.170 μM [1.83 MoM] and/or C16:1-OH ≥ 0.05 μM [3.13 MoM]) were detected in 381 DBS samples. ARSA activity was determined in 230 of the 381 DBS samples (after technical difficulties were resolved) and for 20 DBS samples an ARSA activity below the cutoff (<0.015 μmol/L/h) was determined. Ethical considerations precluded recalling families to provide a second DBS sample for ARSA enzyme testing if insufficient material was available. Because availability of the second-tier ARSA assay was initially delayed, all first-tier screen positives (*n* = 381) received next generation sequencing (NGS) that included the *ARSA* gene as well as *SUMF1* and *PSAP*. Three of the 20 samples with low ARSA activity were confirmed as screen positive for MLD with NGS. Genetic sequencing revealed seven samples with single disease-causing variants in *SUMF1* or *PSAP*, six samples with a single disease-causing *ARSA* variant (considered as being from carriers), and three samples with two, presumably biallelic pathogenic *ARSA* variants. The diagnosis of MLD for all three newborns identified as screen positives for MLD NBS was later confirmed at the qualified treatment center (as described below). In this prospective pilot study, no false positives were reported resulting in a positive predictive value of 100% for the three-tier screening [30]. The pilot study is ongoing and has expanded to Austria [77].

In Italy, an ongoing two-tier prospective pilot NBS program for MLD included the quantification of four sulfatides (C16:0, C16:1-OH, C16:0-OH, and C16:1) as a first-tier test using DBS samples from 42,262 newborns, followed by analysis of ARSA activity for samples with elevated sulfatides as second tier [31]. Sulfatide cutoffs were initially set at the 99.9th percentile based on 5500 healthy newborn samples as follows: C16:0 ≥ 0.196 μM, C16:0-OH ≥ 0.228 μM, C16:1 ≥ 0.041 μM, C16:1-OH ≥ 0.060 μM, and sum of the four sulfatides ≥ 0.485 μM. ARSA activity was determined in 90 (0.21%) samples with elevated levels of one or more sulfatides. In total, 10 newborns (0.02%) were recalled to provide an additional DBS sample because of low ARSA activity (<20% of the daily controls; *n* = 6) or because of insufficient material for a second-tier test (*n* = 4). In all newborns recalled for a new DBS sample, ARSA activity was normal. A retrospective analysis of eight neonatal and 15 non-neonatal DBS samples from patients with MLD was conducted. All samples were positive in the first-tier sulfatide assay, and all 15 samples had undetectable ARSA activity. Neonatal DBS samples from patients with MLD could not be tested for ARSA activity assay validation, owing to known enzyme degradation when stored at room temperature [34,43]. Given the small number of first-tier screen positive cases, an updated cutoff value was assigned for ongoing screening at the 99.0th percentile to enhance the sensitivity of the test, which resulted in the rise in first-tier screen positives from 0.21% to 0.72% after November 2024. This study is ongoing with plans to screen a total of 80,000 newborns.

### 3.3. Confirmatory Diagnostics and Clinical Care Pathways

The studies described above [29,31,32,34] have proven the technical feasibility of multi-tier NBS to identify newborns affected by MLD. Additionally, one prospective study [30] has demonstrated the implementation of a comprehensive clinical care pathway according to European consensus recommendations [11], resulting in all newly identified newborns with early-onset MLD receiving arsa-cel in a timely manner. Newborns with late-onset MLD were referred to a monitoring program so that HSCT could be scheduled before the onset of clinical symptoms [11,30].

The recommended confirmatory diagnostics for MLD screen-positive newborns include leukocyte ARSA activity, preferably determined using a standardized assay with increased sensitivity [9], urinary sulfatides, and genotyping (including inheritance patterns for *ARSA* variants) [11,34]. An improved method for semi-quantitation of urinary sulfatides in patients with MLD has recently been described [63]. Expert consensus recommendations state that infants should be seen for confirmatory testing within two weeks of the NBS positive result for MLD [63]. By using family history (if available), ARSA genotype, and a sensitive assay of ARSA activity, the newborn’s MLD subtype and age of onset can be predicted to inform appropriate clinical care and monitoring [10]. It is important to note that the ARSA activity obtained in the DBS screening assay does not help predict age of onset [11]. Data from publicly accessible variant databases, including ClinVar, Leiden Open Variation Database (LOVD), and Human Gene Mutation Database (HGMD), as well as a comprehensive review of the literature describing MLD cases with identical or functionally equivalent genotypes illustrate that the *ARSA* genotype is a key determinant in estimating the anticipated age of symptom onset. Residual leukocyte ARSA activity correlates with the age at clinical onset, particularly in early-onset phenotypes; hence, its precise quantification is imperative for accurate subtype stratification [9]. In parallel, a thorough neuropediatric assessment is recommended, including a detailed neurological examination, cerebral magnetic resonance imaging (MRI), neurophysiological studies, and gallbladder ultrasonography. Based on the integrated clinical, biochemical, genetic and radiological data, an expert assessment is required to provide evidence-based recommendations for therapy and monitoring [11].

Management for newborns with MLD varies based on MLD subtypes and should be accompanied by genetic counseling, particularly in regard to potentially affected siblings. MLD experts strongly agree that treatment should be provided in the presymptomatic disease stage (with arsa-cel for early-onset MLD subtypes and HSCT for late-onset MLD) [11,17]. Monitoring and management recommendations are available for all MLD subtypes, with detailed monitoring recommendations for late or uncertain disease onset

In a real-world setting, the care pathway has proven effective in 4 screen-positive cases identified to date [30]. Patient data were reviewed with international experts from the MLD initiative (MLDi) [28] to reach a unanimous decision on predicted MLD subtype and the corresponding treatment and monitoring plan. Two newborns with predicted EJ-MLD received arsa-cel at 12 months. The third infant, predicted to have late-onset MLD, is being monitored and is scheduled for HSCT at a later age. A fourth newborn was identified by NBS during submission and predicted to suffer from late infantile MLD. All four children identified were reported to have reached all appropriate developmental milestones [30].

### 3.4. Scalability and Cost-Effectiveness of NBS for MLD

More recently, ScreenPlus, a US NBS pilot program, has been initiated and is aimed at generating real-world data on the appropriateness and feasibility of multi-tiered testing for numerous rare diseases to inform their inclusion in the US RUSP [33]. Following early results from ScreenPlus, in which more than 21,500 infants had been screened as of August 2024, the New York State Department of Health has implemented a state-wide multi-tiered NBS pilot study for MLD [5,35].

A cost-effectiveness study to evaluate adding a three-tier algorithm for MLD to the routine NBS program in the UK calculated the total annual cost of screening 704,328 newborns to be £122,625 (£0.17 per newborn) [46]. In this analysis, using a decision-tree framework, NBS remained cost-effective under a wide range of factors, including fluctuations in MLD incidence and screening test costs. Most importantly, model results showed that only approximately 2.6 of the 7 expected annual MLD cases would be identified early enough to receive arsa-cel treatment with usual case detection. With NBS, however, all cases are expected to be identified in time for presymptomatic treatment, leading to significantly improved survival and quality of life for newborns with MLD [46]. Similar results were reported in two congress abstracts related to the cost-effectiveness of screening for MLD in the US [69,70]. Using a similar decision tree model framework, one of the studies showed that screening for MLD in California would result in a positive net economic benefit of more than USD 70 million and the avoidance of 5.09 premature deaths [69]. Overall, these results from these three cost-effectiveness studies support the inclusion of MLD into routine NBS programs [70].

### 3.5. Benefits of Early Treatment

A total of 11 publications reporting outcomes of patients with MLD were identified and are summarized in Table 2. Of these, five reported outcomes in patients receiving arsa-cel [15,47,48,49,71] and five reported outcomes in patients treated with HSCT [23,24,25,26,72]. None of the publications reported quality of life outcomes. Outcomes of treatment with arsa-cel and HSCT were also compiled in a recent systematic literature review [27].

#### 3.5.1. Arsa-Cel Treatment

Benefits of early treatment with arsa-cel on motor and cognitive function have recently been reported in detail in three key publications [15,47,48] with overlapping groups of patients, including data from two prospective open-label clinical studies (NCT03392987 and NCT01560182) and expanded-access programs.

The first publication compared long-term outcomes (up to 7.5 years of follow-up) in 29 pre-symptomatic or early-symptomatic patients with MLD (pre-symptomatic LI, *n* = 16; pre-symptomatic EJ, *n* = 5; and early-symptomatic EJ, *n* = 8) treated with arsa-cel with the outcomes of patients in an untreated natural history cohort (*n* = 31) [47]. A more recent integrated analysis of those patients treated with arsa-cel and others (a total of 37 patients) reported efficacy and safety findings over a longer period (up to 12.2 years of follow-up; median: 6.76 years) [15]. The adverse events reported were consistent with the known safety profile of the busulfan conditioning regimen used prior to arsa-cel infusion or with symptoms of MLD. Outcomes in patients with pre-symptomatic LI-MLD (*n* = 18), pre-symptomatic EJ-MLD (*n* = 8) and early-symptomatic EJ-MLD (*n* = 11) were compared with outcomes in patients from the same natural history cohort. In line with previous findings, arsa-cel treatment was associated with a significantly lower risk of all-cause death or severe motor impairment defined as loss of locomotion and sitting without support (Gross Motor Function Classification in MLD [GMFC-MLD] Level 5) than no treatment in patients with pre-symptomatic LI-MLD (*p* < 0.001), pre-symptomatic EJ-MLD (*p* = 0.04) and early-symptomatic EJ-MLD (*p* < 0.001). Motor function, cognitive function, and language skills were better maintained in patients with presymptomatic LI-MLD who were treated with arsa-cel than in those with untreated LI-MLD. All surviving patients with presymptomatic EJ-MLD who were treated with arsa-cel had age-appropriate motor, cognitive, and language skills, with five out of seven patients having surpassed the age at which the onset of symptoms had occurred in their untreated sibling and four of seven having surpassed the median age at which untreated patients with EJ-MLD enter GMFC-MLD Level 5 [15]. Outcomes in patients with early-symptomatic EJ-MLD were more variable. Gross motor function was better preserved in patients with EJ-MLD who were treated when they were pre-symptomatic than those who were treated after they had developed early symptoms [15,47]. The clinical trial data [15,47] and real-world experience in evaluating eligibility for treatment [16,28] underscore the need to determine if a patient with LI-MLD has become symptomatic or whether a patient with early-symptomatic EJ-MLD has entered a phase of rapid disease progression, because such patients do not appear to benefit from arsa-cel.

A post hoc analysis of 15 patients with pre-symptomatic LI-MLD patients treated with arsa-cel (included in the studies by Fumagalli et al. described above [15,47]) assessed the efficacy of arsa-cel in mitigating the severity and progression of peripheral neuropathy as assessed by nerve conduction velocity (NCV) [48]. At 2 years post-treatment, NCV was significantly higher in treated patients than in untreated patients (Table 2). Younger age at treatment was associated with increased NCV in the ulnar and median nerves [48].

Two case reports of single patients with pre-symptomatic LI-MLD treated with arsa-cel were identified. In the first one [49], the patient received arsa-cel at 15 months as part of the trials described by Fumagalli et al. [15,47]. In the second one [71], the patient received arsa-cel at 11 months. Both reports highlighted the benefits of early arsa-cel treatment, reporting preserved motor function and cognitive skills (relative to what would be expected in an untreated patient).

Fahim et al. 2024 [52] assessed the cost-effectiveness of arsa-cel treatment compared to usual care. The treatment was significantly more cost-effective for patients with pre-symptomatic LI-MLD and pre-symptomatic EJ-MLD than for those with early-symptomatic EJ-MLD due to the marked difference in achievable outcomes for presymptomatic vs. early symptomatic treated patients.

#### 3.5.2. Allogeneic Hematopoietic Stem Cell Transplantation

Several observational studies have reported long-term outcomes following treatment with HSCT [23,24,25,26,72]. In these studies, an older classification of juvenile MLD (J-MLD) was used, which did not consistently distinguish between EJ-MLD and LJ-MLD.

A retrospective cohort study of 40 patients with MLD (LI-MLD, *n* = 4; J-MLD, *n* = 27 and adult MLD [A-MLD], *n* = 9) treated with HSCT in the US showed a 5-year survival of 59% in the overall population and was independent of MLD subtype and presence of symptoms at the time of transplantation [24]. Most patients who underwent transplantation showed adaptive behavior functional decline during follow-up. Although the study did not include a control group, five sibling pairs (one LI-MLD and four J-MLD), in which at least one sibling underwent transplantation, were evaluated. Within each pair, survival or function was superior for the treated sibling, or if both siblings had undergone transplantation, for the pre-symptomatic sibling.

Another retrospective study compared long-term outcomes (median follow-up: 7.5 years) in 24 patients with J-MLD treated with HSCT and 41 untreated patients with J-MLD [25]. Survival following disease onset was similar in treated and untreated patients, but HSCT slowed the decline of gross motor and cognitive function. An exploratory analysis identified potential predictors of stable disease: GMFC-MLD levels 0 or 1 (vs. higher levels) at time of HSCT, IQ ≥ 85 at time of HSCT, and age at disease onset > 4 years. Conversely, brain MRI severity scores > 17 were associated with disease progression after HSCT.

An analysis of observational data from the Netherlands compared outcomes in 13 patients with MLD who received HSCT (LI-MLD, *n* = 2; J-MLD, *n* = 5; A-MLD, *n* = 6) with a group of 22 patients who did not receive HSCT [26]. Overall survival at last assessment was similar in the treated and untreated groups (76.9% vs. 63.6%; *p* = 0.62). Intervention-free survival (IFS) and activities of daily living compromise-free survival (AFS) were, however, significantly higher in patients who received HSCT than in those who did not. In treated patients, IFS and AFS were numerically greater for patients who were asymptomatic at the time of HSCT than for those who were symptomatic.

A small retrospective study of 12 patients with J-MLD who received HSCT reported varying short-term outcomes (2 years post-treatment) [23]. Seven patients remained stable (exhibited no, or only mild deterioration of motor function; ≤1 level of GMFC-MLD), and five patients deteriorated in the first 12–18 months after transplantation (>1 level of GMFC-MLD). Cognitive decline mirrored motor function deterioration. The authors noted that transplanted patients with disease progression had abnormal gross motor function at the time of HSCT, and concluded that transplantation should take place early, before gross motor symptoms appear.

One congress abstract [72] reported outcomes for patients with LI-MLD who received HSCT that differ from the majority of published studies. HSCT, even in asymptomatic patients, does not represent the standard of care for patients with early-onset MLD [11,24,27,28,78,79].

### 3.6. Summary

Given the availability of effective treatments in the presymptomatic stage and the feasibility of DBS-based screening, experts worldwide strongly support implementing NBS programs for MLD [11,17]. The inclusion of MLD in NBS programs is based on historical objective criteria [80], including accurate confirmatory diagnostics and early treatment strategies that improve clinical outcomes [16,45].

## 4. Discussion

This systematic review provides a comprehensive summary of the evidence indicating that MLD is suitable for incorporation into NBS programs (see Figure 2). NBS for MLD fulfills all of the classical Wilson and Jungner principles [80]. The evidence includes the availability of a reliable screening algorithm, confirmatory diagnostics, data on subtype prediction, and a clinical care pathway for all MLD subtypes. From a public health perspective, the ability to identify affected infants in a population-based, cost-effective NBS program combined with the availability of an approved gene therapy for early-onset MLD and HSCT for late-onset highlights the critical importance of pre-symptomatic diagnosis as opposed to waiting for clinical presentation [16,45,81].

### 4.1. Screening Assays

The development of screening assays for MLD has followed a pathway similar to other NBS programs: initial assay design, refinement of cutoff values with insights from pilot studies, algorithm optimization, and prospective validation. Early approaches anticipated that there would be challenges in using ARSA enzyme activity as a reliable first-tier marker due to ARSA pseudodeficiency alleles and carriers, and the instability of the enzyme in DBS samples stored at room temperature [1,8,34,43]. These limitations prompted exploration of an antibody-based method to measure the concentration of ARSA protein [68] and ultimately led to the successful adoption of sulfatide measurement in DBS samples as a first-tier test [42]. Progress is reflected in the refinement of the DBS screening algorithm and the cutoffs first utilized by Hong et al. 2021 [29]. The need for refinement of cutoff values is exemplified by Wu et al. 2024 [34], who reported a case of LI-MLD with a ‘false negative’ sulfatide result. It should be noted that the individual screening tests, cutoff values, and MLD NBS algorithm were still being validated at the time that this case was identified [34], which, therefore, was not a false negative in the true sense of the term in relation to NBS for MLD.

NBS for MLD now follows a robust three-tiered screening algorithm including first-tier testing of sulfatides, second-tier testing of ARSA activity and third-tier sequencing of the *ARSA* gene (see Figure 3). Utilization of all three tiers is essential to ensure optimal screen positive and patient recall rates, as demonstrated in a population-based program [30,31].

Optimizing sulfatide species analysis and establishing appropriate cutoffs for sulfatide levels that balance test sensitivity and specificity will be important considerations for scalability and the implementation of MLD screening across NBS programs. Sulfatide levels in DBS from healthy infants are known to increase with age within the first 2 years of life [32,34]. Therefore, NBS laboratories will need to take this into account when setting cutoffs. In addition, using MoM instead of fixed cutoffs can improve inter-laboratory consistency and enables more reliable benchmarking, which is especially important given the rarity of positive samples available for validation [34]. ARSA activity in DBS is temperature sensitive and degrades over time, which underscores (i) the need to establish normal population ranges and cutoff values for each NBS program and (ii) the importance of the third-tier sequencing of the ARSA gene to avoid unnecessary recall of neonates [62]. Although DBS samples can be stored under conditions that mitigate activity loss [34,43], utilizing ARSA activity as a second-tier assay for MLD screening is expected to reduce challenges commonly observed in other LSD screening approaches, where using enzyme activity as a first-tier test often leads to positive screens driven by temperature-related variability [62,82].

Collaboration among screening laboratories, as exemplified in several publications [32,62], is essential for the successful implementation of future MLD screening programs, as it enables robust and efficient assay validation despite the extremely limited number of appropriately stored DBS samples from newborns who are later confirmed to have MLD [30,31,32]. Additionally, systematic analysis of DBS samples from MLD carriers and individuals with ARSA pseudodeficiency that are collected during the newborn period will be necessary to confirm or refine cutoff values and maximize performance of the NBS algorithm in future MLD NBS programs.

### 4.2. Confirmatory Diagnostics and Prediction of Subtype

Following a positive screen, international consensus publications provide structured guidance on clinical assessments to confirm diagnosis and predict MLD subtype, which inform monitoring strategies and care pathways [11,17,30]. This was explicitly demonstrated in a real-world scenario with identification and successful treatment of three patients with MLD in the German pilot study [30]. Genetic databases, genotype–phenotype data and results from a modified leukocyte ARSA activity [9] support the prediction of MLD subtype in the majority of cases, and the consensus recommendations address cases in which subtype classification may remain uncertain [11,17]. With the predicted expansion of NBS for MLD, the number of identified variants of uncertain significance (VUS) in the *ARSA* gene will increase, as has been shown with NBS program for other metabolic disorders [83]. In vitro assays and cell expression platforms capable of defining the severity and functional consequences of such variants [10,83,84] have been described and will assist providers in their ability to make an accurate prediction about MLD subtype following a positive screen. The rapidly evolving field of in silico modeling may also prove helpful in assessing the impact of a novel *ARSA* variant on enzymatic activity [85,86].

### 4.3. Prospective, Population-Based NBS Programs for MLD

Two prospective pilot studies have been conducted to date, in Germany [30] and in Italy [31]. In both studies, implementation of second-tier ARSA activity testing was not fully optimized at the time of study initiation. Nevertheless, findings from these and other studies [29,32,34] reveal performance characteristics that reinforce the efficacy of the optimized three-tier algorithm and provide a strong foundation for broader implementation. Screening for MLD may be multiplexed with tests for other inborn errors of metabolism that are already deployed in NBS centers worldwide [32,33].

Moreover, the results of these studies indicate that the number of infants recalled for diagnostic testing who are not ultimately confirmed to have MLD (i.e., false positives based on three-tier testing) will be close to zero, thereby minimizing unnecessary psychosocial burden on families and costs to the healthcare systems. While long-term follow-up data will facilitate further refinement of screening pathways, absence of these data should not delay broader implementation of MLD NBS.

Several MLD NBS pilot studies are ongoing, and MLD has been added or nominated for inclusion in NBS programs around the world [30,31,32,33,34,35,37,38,39,40]. Following the publication of the first prospective NBS pilot for MLD, Norway was the first country to implement nationwide screening for MLD in January 2025 [40]. In the absence of commercially available test kits, Norway has implemented NBS using laboratory-developed test methodology.

Several genomic NBS programs are currently underway that use NGS to screen newborns for known genotypes implicated in pediatric diseases, including MLD, but the role of this analytical approach in population-based NBS has not yet been established [87,88]. For the foreseeable future, biochemical-based screening for MLD, as described in the results, will remain the predominant method. Importantly, biochemical screening functions as an equalizer in public healthcare systems, because it circumvents the limitations imposed by genetic heterogeneity and the underrepresentation of certain ethnic groups in public variant databases. The biochemical approach enables the unbiased identification of affected newborns, regardless of their genetic background or the presence of private *ARSA* variants, complex *ARSA* genotypes or structural variants that may be missed by sequencing depending on the method and bioinformatic analyses utilized [88].

### 4.4. Improved Health Outcomes from Early Identification

Both arsa-cel and allogeneic HSCT have limited or no effectiveness in patients with MLD who are symptomatic or who have entered the rapidly progressive phase of their disease [15,23,24,25,47]. This is because the mechanism of action of both treatments requires sufficient time for hematopoietic cells to engraft and for their progeny to migrate and produce ARSA in the nervous system, a process that is insufficient to counteract the speed at which irreversible neurophysiological deterioration occurs in such patients.

Disease progression in untreated patients with the LI and EJ forms of MLD is highly predictable, with fast deterioration in motor and cognitive function occurring within months once patients enter the rapidly progressive phase of the disease [6,7,89]. As with other pediatric neurodegenerative diseases, treatment of patients with early-onset MLD with arsa-cel as early as possible will provide the greatest clinical benefit, given the narrow window to intervene before serious and irreversible progression occurs. The importance of early diagnosis and treatment of MLD and the vital role of NBS in this paradigm is stressed in recent expert consensus guidelines [11,17].

There are unequivocal clinical trial data showing that patients treated in the pre-symptomatic period have the best long-term outcomes, including results with arsa-cel for early-onset MLD and HSCT for late-onset MLD [15,23,24,25,47]. A recent report suggesting the benefit of autologous HSC gene therapy patients with advanced symptomatic late-onset MLD [90] has been discredited [91]. International consensus recommendations advise that patients with pre-symptomatic early-onset MLD receive arsa-cel treatment within the first year of life [11,17]. Arsa-cel is not currently approved for the treatment of late-onset MLD, but there is an ongoing clinical study of arsa-cel in patients with LJ-MLD (NCT04283227) [92].

Cellular and organ damage attributable to toxic sulfatide accumulation in MLD starts very early in the disease, including in infants who are considered clinically pre-symptomatic. The abnormal sulfatide accumulation that is detected in newborns can also be detected prenatally [93,94]. Clinical evidence of disease due to progressive pathology before the overt onset of neurological symptoms includes abnormal NCV, gallbladder abnormalities and other prodromal findings in pre-symptomatic infants with MLD, and abnormal brain MRIs in some patients with clinically pre-symptomatic EJ-MLD [25,30,48]. Although patients with mild symptoms may still benefit from treatment, the presence of biochemical, clinical, or subclinical indicators of disease prior to overt symptom onset alongside the inexorably progressive and irreversible nature of untreated MLD strongly support the need for identification and intervention as early as possible.

Results from the arsa-cel clinical development program demonstrate the benefit of early treatment for LI-MLD and EJ-MLD, with the best outcomes observed in patients treated prior to symptom onset [15]. This is particularly apparent in the difference in outcomes between pre-symptomatic and early-symptomatic patients with EJ-MLD. Although patients with early-symptomatic EJ-MLD benefit from treatment and experience stabilization or slower decline in motor function compared with untreated patients, outcomes in patients with early-symptomatic EJ MLD were much more variable than those who were treated when pre-symptomatic [15]. The fact that every surviving pre-symptomatic patient with EJ-MLD had normal motor and cognitive function at their last follow up highlights the impact of presymptomatic treatment. The effect of arsa-cel treatment on improved health-related quality of life in early-onset MLD patients has been reported in a global survey study of caregivers [95].

The absence of NBS for MLD raises serious ethical concerns because it results in missed opportunities for timely intervention despite the availability of an effective therapy to prevent disease progression. This is reflected in data from the largest cohort of patients with early-onset MLD treated with arsa-cel in a real-world setting, in which only four out of 17 referred patients could be treated. Thirteen patients were deemed ineligible for treatment, including 10 with LI-MLD who were already symptomatic at evaluation [16]. In the absence of a family history, most patients are currently identified too late for effective intervention [56]. Given the nonspecific nature of many early disease manifestations and the likelihood that subclinical features go unnoticed or are misinterpreted by both parents and pediatricians, clinical recognition of MLD early enough for most such patients to benefit from therapy is highly unlikely. Implementing NBS offers a critical opportunity to change this trajectory by enabling diagnosis at birth and treatment before irreversible neurological damage occurs.

### 4.5. Balancing Harm and Benefit of NBS for MLD

A robust NBS algorithm has been developed for the screening for MLD. However, as it is a measure of increased risk of disease and not a diagnostic test, false positive recalls are possible. Concerns about the risk of NBS doing harm—particularly around false positives or prognostic uncertainty—are not unique to MLD, and data across a range of conditions consistently suggest that the actual harm is more dependent on the context and quality of result disclosure than the result itself [44]. Studies have emphasized the importance of effective communication and the availability of psychosocial support to mitigate distress in the event of false positive recall [96,97,98].

Findings from multiple caregiver studies reinforce the value and perceived benefit of early diagnosis. In a recent caregiver survey from the UK and Republic of Ireland, 85% of respondents indicated that NBS for MLD would have influenced their reproductive decision-making, and 86% believed that earlier detection would have changed their child’s future [56]. Notably, 80% considered missing a diagnosis at birth to be more harmful than receiving a false positive result, emphasizing how highly families value early knowledge and intervention opportunities.

Supporting these conclusions, a study of parents of children diagnosed with LSDs through NBS showed that early diagnosis via screening was associated with significantly lower odds of parental depression compared with diagnosis through clinical presentation or family history [99]. These results challenge longstanding assumptions about the psychological risks of expanded NBS and instead underscore the protective effect of early, structured intervention pathways.

### 4.6. Limitations

This systematic review has several limitations. The type of methodological diversity observed across the studies of MLD screening assays (including differences in sulfatide species measured, cut-off thresholds, number of screening tiers, and the use of genetic testing) is expected during the early development of a novel NBS algorithm, but it complicates cross-study comparison, introduces variability in reported performance metrics and precludes formal meta-analysis at this time. Most evidence comes from a relatively small number of observational and retrospective investigations, small-scale feasibility studies, and early pilot programs, with only a limited number of screen-positive infants reported to date. Conducting a randomized controlled trial for a rapidly progressing, fatal rare disorder with no existing targeted treatment, like MLD, was neither feasible nor ethical. Additionally, the studies of outcomes of treatment with arsa-cel and HSCT utilized appropriate external natural history patient comparator groups, an approach that is accepted by health authorities.

Potential conflicts of interest include acknowledgment that several co-authors of this review are associated with Orchard Therapeutics, the developer of arsa-cel, and/or have co-authored some of the studies included in the evidence base. Such overlap reflects the concentration of expertise within the small international community typically working on a rare disease like MLD. While these connections have the potential to introduce interpretive bias, the risk was mitigated by conducting data extraction and interpretation with independent academic and clinical co-authors and critical appraisal by a multidisciplinary team.

### 4.7. Future Perspectives

The current MLD NBS algorithm shows promise, comparing favorably with programs for other LSDs at a similar stage of development [44,76,100]. Progress has accelerated in recent years, owing to methodological refinements and lessons learned from two decades of experience in other LSD screening initiatives.

Genotype–phenotype correlations and measures of residual ARSA activity will increasingly guide prognosis. Ongoing contributions to public databases and collaborative analyses of MLD cases, including those with VUS and previously unreported genotypes, will be essential to broaden knowledge and improve clinical interpretation.

Future research must prioritize the collection of comprehensive real-world data across diverse healthcare systems. Prospective studies should capture not only clinical endpoints such as motor, cognitive, and survival outcomes, but also patient- and family-reported experiences, including psychosocial impact, healthcare use, and quality of life.

As programs expand, sustained follow-up systematic evaluation will help refine algorithms, reduce false positive and false negative results, and mitigate risks linked to prolonged monitoring of children with unclear prognoses. Broader participation in international registries and harmonized data-sharing frameworks will be critical for enabling cross-program comparisons, accelerating evidence generation, and supporting consensus on best practices. As empirical evidence accrues, economic evaluations should be revised as evidence builds on treatment durability, long-term benefit, and healthcare resource use.

MLD NBS is expected to deliver meaningful clinical benefit. To maximize its impact, widespread adoption should be accompanied by ongoing research that continues to strengthen its scientific, clinical, ethical, and economic foundations.

## 5. Conclusions

This review provides the first comprehensive summary of the published evidence on NBS in MLD and has been written to provide clarity and structure from the emerging data as well as an evidence base from which policy decisions and future work can build. The findings may support groups who are nominating MLD for inclusion on screening panels and committees who are reviewing the available evidence.

Assays for a multi-tier screening algorithm for MLD NBS are available. Data indicate that the algorithm is sensitive and specific for MLD, and consensus on protocols and care pathways have been developed. This multi-tier testing has been validated and applied in prospective MLD NBS pilot studies to detect patients with pre-symptomatic MLD and enable their timely treatment.

## Figures and Tables

**Figure 1 IJNS-11-00103-f001:**
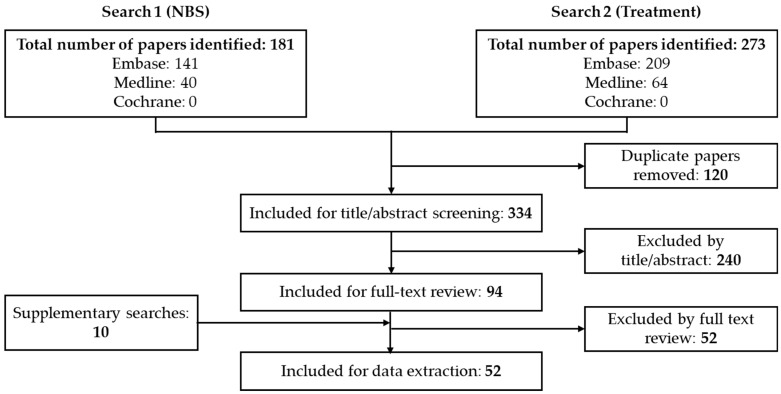
PRISMA flow diagram.

**Figure 2 IJNS-11-00103-f002:**
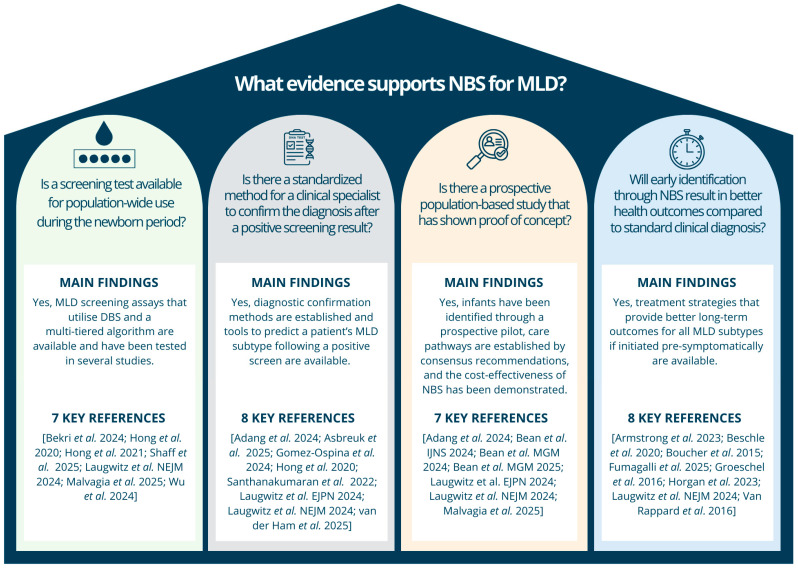
Summary of Key Supporting Evidence. [1,2,9,11,15,16,17,23,24,25,26,27,29,30,31,32,34,43,46,62,63,69,70].

**Figure 3 IJNS-11-00103-f003:**
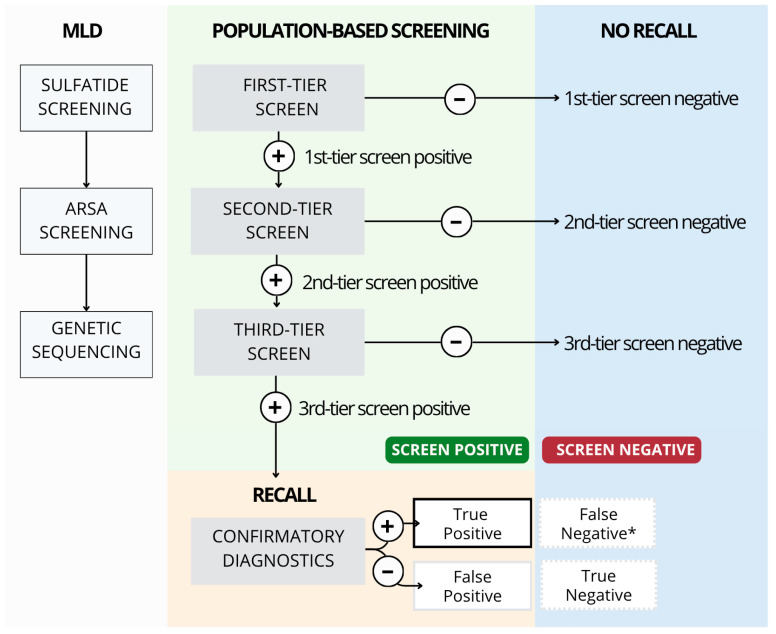
Three-tiered Population-Based Screening Algorithm Terminology. * Because infants with a screen negative result are not recalled, they could only be classified as a false negative if they were later diagnosed with MLD due to clinical symptoms.

**Table 1 IJNS-11-00103-t001:** An overview of MLD newborn screening studies.

	Method Development & Feasibility Studies			Prospective Pilot Studies	
	**Hong et al. (2021)** [29]	**Wu et al. (2024)** [34]	**Bekri et al. (2024)** [32]	**Laugwitz et al. (2024)** [30]	**Malvagia et al. (2025)** [31]
**Country**	US	UK	Multiple	Germany	Italy
**Study design**	Retrospective pilot	Retrospective pre-pilot	Retrospective analysis	Prospective pilot	Prospective pilot
**First-tier screen**	C16:0 (≥0.170 μM)	C16:0 (≥0.170 μM)	C16:0 (1.65 MoM) and/or C16:1-OH (2.70 MoM)	C16:0 (≥0.17 μM) and/or C16:1-OH (≥0.05 μM)	C16:0 (≥0.196 μM), C16:0-OH (≥0.228 μM), C16:1 (≥0.041 μM), C16:1-OH (≥0.060 μM) and sum of the four sulfatides (≥0.485 μM)
**Second-tier screen**	ARSA activity (<20% of daily mean)	ARSA activity (<20% of daily mean)	NA	ARSA activity (≤0.015 μM/h)	ARSA activity (<20% of daily mean)
**Third-tier screen**	*ARSA* gene	*ARSA* gene	NA	*ARSA*, *SUMF1*, *PSAP* genes	*ARSA* gene
**Genetic sequencing technique**	NGS	Sanger sequencing	NA	NGS	NR
**Number of newborns screened**	27,335	3687	135,824 ^c^	109,259	42,262
**Number of first-tier** **positives**	195 (0.71%)	11 (0.30%)	C16:0: 2456 (1.8%) C16:1-OH: 64 (0.048%) (0.030%)	381 (0.35%)	90 (0.21%) ^f^
**Number of second-tier positives**	2 ^a^	0	NA	20 ^d^	6 ^g^
**Number of third-tier positives**	1	NA	NA	3 ^e^	NA
**Confirmed MLD cases**	None ^b^	1	NR	3	0
**MLD samples for first-tier assay validation**	15/15 detected	NR	40/40 detected	Cross-validated with Bekri et al. 2024 [32]	23/23 detected
**Limitations/Challenges**	Unable to recall the infant with a screen positive sample; only 122/195 samples with elevated sulfatides available for second-tier assay due to DBS storage time	Special review by UK research and health authorities was needed to obtain permission to de-anonymize a pre-pilot DBS sample	Study designed to evaluate the first-tier assay only	ARSA assay not available until the study was already underway	High recall rate due to absence of a third-tier screening ^g^, second DBS sample required for genetic sequencing

^a^ Only 122 samples out of 195 with elevated sulfatides were submitted for ARSA activity; the other 73 samples were not tested because the DBS samples were too old (stored for more than 3 months at room temperature). Therefore, the percentage of second-tier positives could not be calculated. ^b^ The third-tier positive sample could not be confirmed as an MLD case because the study was retrospective and used de-identified DBS samples. ^c^ Includes 592 samples assessed at the University of Washington, 3687 at the Manchester Biochemical Genetics Laboratory, 5000 at the Rouen University Hospital and 126,545 at ARCHIMEDlife. ^d^ Owing to early technical challenges and storage time, only 230 of the 381 samples with elevated sulfatides were analyzed for ARSA activity. Therefore, the percentage of second-tier positives could not be calculated. ^e^ Genetic testing was conducted on all 381 DBS samples with elevated sulfatides. ^f^ Updated cutoff values after November 2024 resulted in first-tier screen positives of 0.72%. ^g^ Six newborns with low ARSA activity and four newborns who had insufficient residual material for the second-tier assay were recalled. ARSA activity was normal in all 10 new samples.

**Table 2 IJNS-11-00103-t002:** Key findings from studies reporting treatment outcomes.

Study	Study Type	Numbers of Patients	Disease Subtype (Treated Group)	Follow-Up	Key Outcomes and Study Findings
**Arsa-cel**					
Fumagalli et al. 2022 [47] ^a^	Analysis of data from two prospective non-randomized, open-label, phase 1/2 clinical study and expanded-access frameworks	29 (arsa-cel) 31 (NHx)	PS LI-MLD (*n* = 16) ^b^ PS EJ-MLD (*n* = 5) ES EJ-MLD (*n* = 8)	Median FU (range), in yrs: Overall: 3.16 (0.64–7.51) LI-MLD: 3.04 (0·99–7.51) EJ-MLD: 3.49 (0.64–6.55)	Motor function at 2 years (mean differences in total GMFM-88 scores between treated patients and age-matched and MLD subtype-matched untreated patients) LI-MLD: 66% (95% CI: 48.9–82.3) EJ-MLD: 42% (95% CI: 12.3–71.8) Cognitive function throughout FU Remained normal in 80% of assessed patients receiving arsa-cel (20/25) NHx patients showed severe deficits Severe motor impairment-free survival (treated patients vs. MLD subtype-matched untreated patients) PS LI-MLD: 92% (95% CI: 57–99) vs. 0% at 4.5 yrs PS EJ-MLD: 80% (95% CI: 20–97) vs. 36% (95% CI: 9–60) at 8 yrs ES EJ-MLD: 63% (95% CI: 23–86) vs. 36% (95% CI: 9–60) at 8 yrs
Fumagalli et al. 2025 [15] ^a^	Analysis of data from two prospective non-randomized, open-label, phase 1/2 clinical study and expanded-access frameworks	37 (arsa-cel) ^c^ 49 (NHx)	PS LI-MLD (*n* = 18) PS EJ-MLD (*n* = 8) ES EJ-MLD (*n* = 11) ^d^	Median FU (range), in yrs: PS LI-MLD: 6.7 (2.4–12.2) PS EJ-MLD: 3.8 (1.1–9.6) ES EJ-MLD: 7.4 (0.6–9.4)	Primary efficacy endpoint Arsa-cel resulted in a significantly lower risk of severe motor impairment (defined as GMFC-MLD level ≥ 5) or death than no treatment PS LI-MLD: *p* < 0.001 PS EJ-MLD: *p* = 0.04 ES EJ-MLD: *p* < 0.001 Estimated percentages of patients surviving without severe motor impairment (defined as GMFC-MLD level ≥ 5; treated patients vs. age-matched and MLD subtype-matched untreated patients) PS LI-MLD: 100% (95% CI: 100–100) vs. 0% (95%CI not evaluable) at 6 yrs of age PS EJ-MLD: 87.5% (95% CI: 38.7–98.1) vs. 11.2 (95% CI: 0.9–36.4) at 10 yrs of age ES EJ-MLD: 80.0% (95% CI: 40.9–94.6) vs. 11.2 (95% CI: 0.9–36.4) at 10 yrs of age Percentages of patients surviving without severe motor impairment at 2 yrs after treatment (defined as GMFC-MLD level ≥ 5; treated patients vs. age-matched and MLD subtype-matched untreated patients). PS LI-MLD: 100% (95% CI: 81–100) vs. 40% (95% CI: 21–61); *p* < 0.001 PS EJ-MLD: 88% (95% CI: 47–100) vs. 100% (95% CI: 78–100); not significant ES EJ-MLD: 82% (95% CI: 48–98) vs. 87% (95% CI: 60–98); not significant Overall survival Significantly higher in treated patients with PS LI-MLD vs. untreated patients with LI-MLD (*p* < 0.001); survival at 6 years: 100% (95% CI: 100–100) vs. 59.0% (95% CI: 37.2–75.5) Similar in treated patients with PS or ES EJ- patients and untreated patients with EJ-MLD Survival free from motor impairment (defined as GMFC-MLD ≥ 3) (treated patients vs. MLD subtype-matched untreated patients) PS LI-MLD: 93.8% (95% CI: 63.2–99.1) vs. 0% (95% CI not evaluable) at 6 yrs of age PS EJ-MLD: 87.5% (95% CI: 38.7–98.1) vs. 0% (95% CI not evaluable) at 10 yrs of age ES EJ-MLD: 57.1% (95% CI: 21.7–81.5) vs. 0% (95% CI not evaluable) at 10 yrs of age Adjusted mean GMFM-88 scores (treated patients vs. age-matched and MLD subtype-matched untreated patients) At 2 years PS LI-MLD: 79.41 vs. 8.97 PS EJ-MLD: 94.27 vs. 41.92 ES EJ-MLD: 86.88 vs. 39.64 At 5 years PS LI-MLD: 79.94 vs. 1.26 PS EJ-MLD: 99.96 vs. 23.99 ES EJ-MLD: 47.34 vs. 7.48 Estimated percentages of patients surviving without severe cognitive impairment (treated patients vs. age-matched and MLD subtype-matched untreated patients) PS LI-MLD: 100% (95% CI: 100–100) vs. 8.8% (95% CI:1.5–24.3) at 6 yrs of age PS EJ-MLD: 87.5% (95% CI: 38.7–98.1) vs. 7.5 (95% CI: 0.5–28.4) at 10 yrs of age ES EJ-MLD: 64.8% (95% CI: 025.3–87.2) vs. 7.5% (95% CI: 0.5–28.4) at 10 yrs of age
Zambon et al. 2025 [48] ^a^	Post hoc analysis of data from two prospective non-randomized, open-label, phase 1/2 clinical study and expanded-access frameworks	15 (arsa-cel) 16 (NHx)	PS LI-MLD	Median FU (range), in yrs: 4.69 (1.95–12.11)	NCV values at 2 years (treated vs. untreated patients) DPN: 24.5 vs. 10.1 m/s (*p* < 0.0001) UN: 32.1 vs. 14.2 m/s (*p* < 0.0001) MN: 33.2 vs. 1.5 m/s (*p* < 0.0001) SN: not estimated because many values were 0
Faccioli et al. 2023 [49] ^a^	Case report	1	PS LI-MLD	106 months	Patient treated at 15 months of age and stable until the age of 6 years (increasing gait difficulties thereafter)
Faqueti et al. 2023 [71]	Case report	1	PS LI-MLD	4 months	Patient surpassed the age of onset of his older brother Developing well and acquiring new motor and cognitive skills
**HCST**					
Beschle et al. 2020 [23]	Case–control study	12 (HSCT) 35 (NHx)	J-MLD	Mean FU (range), in yrs: 6.75 (3.0–13.5)	Disease progression in treated patients within 2 years of treatment 7 patients remained stable (5 of these patients remained stable throughout the entire observation period; all had GMFC-MLD = 0, GMFM-88 = 100% and MRI severity score ≤ 17 at time of HSCT) 5 patients deteriorated (motor and cognitive functions) Predictive values for disease progression after HSCT (all *p* values were considered as descriptive) GMFM-88 < 100%: *p* = 0.003 GMFC-MLD > 0: *p* = 0.013 FSIQ < 85: *p* = 0.079 MRI severity score > 17: *p* = 0.003 NCV (tibial nerve) < 40 m/s: *p* = 0.198 Age at onset: *p* = 0.690 Age at HSCT: *p* = 0.530
Boucher et al. 2015 [24]	Retrospective cohort study	40	LI-MLD (*n* = 4) J-MLD (*n* = 27) A-MLD (*n* = 9)	Median FU (range), in yr, mos: 10.0 (0.1–30.6)	Overall survival at 5 years Entire cohort: 59% (95% CI: 42–73) LI-MLD: 50% (95% CI: 6–84) J-MLD: 59% (95% CI: 38–75) A-MLD: 67% (95% CI: 28–88) Cognitive and motor function Although all three evaluable patients with LI-MLD demonstrated normal-for-age adaptive behavior at baseline, the lone patient with extensive follow-up showed significant eventual functional decline The majority of evaluable patients with J-MLD or A-MLD demonstrated adaptive behavior impairment at baseline, and most continued to show decline Patients with J-MLD showed better preservation of motor and expressive language function than patients with LI-MLD
Groeschel et al. 2016 [25]	Retrospective cohort study	24	J-MLD	Median FU (range), in yrs: 7.5 (3.0–19.7)	Survival *Treated* 4 transplantation-related deaths 2 patients with rapid disease progression (died 1.5 and 8.6 years after HSCT) 9 long-term survivors had disease progression 11 long-term survivors had stable disease Resulting in 79% survival at 5 yrs post treatment *Untreated* 5-year survival after disease onset was 100% (41 of 41). However, 11 (27%) died of MLD progression, resulting in similar overall survival to treated patients within the observation period Cognitive function Treated patients were less likely to lose any language function after disease onset than untreated patients (*p* = 0.07) Ten years after the first symptoms, language loss was observed in 40% (8 of 20) of the treated patients vs. 68% (28 of 41) of untreated patients Motor function Treated patients were more likely to maintain their gross motor function and not to progress to GMFC-MLD level 5 after disease onset than untreated patients (*p* = 0.04) Prognostic parameters at time of HSCT for stable vs. progressive disease: GMFC-MLD ≤ 1: *p* = 0.02 IQ ≥ 85: *p* = 0.02 MRI severity score ≤ 17: *p* = 0.03 Age at onset > 4 years: *p* = 0.01
Van Rappard et al. 2016 [26]	Prospective longitudinal study	13 (HSCT) 22 (no HSCT)	LI-MLD (*n* = 2) J-MLD (*n* = 5) A-MLD (*n* = 6)	Mean FU, in yrs: HSCT: 4.7 NHx: 4.6	Overall survival at last assessment (HSCT vs. no HSCT): 76.9% vs. 63.6% (*p* = 0.62) Intervention-free survival (no occurrence of death, wheelchair dependency, gastrostomy, or intrathecal baclofen treatment; HSCT vs. no HSCT): 69.2% vs. 9.1% (*p* = 0.03) Activities of daily living compromise-free survival (no occurrence of death, motor [clinically relevant peripheral neuropathy, spasticity, or ataxia, gross motor function ≥ 3], or cognitive [IQ decline ≥ 6 points] deterioration; HSCT vs. no- HSCT): 46.2% vs. 0% (*p* = 0.01) Intervention-free survival in patients who underwent HSCT (pre-symptomatic vs. symptomatic at time of HSCT): 100% vs. 42.9% (*p* = 0.052) Activities of daily living compromise-free survival in patients who underwent HSCT (pre-symptomatic vs. symptomatic at time of HSCT): 66.7% vs. 28.6% (*p* = 0.11)
Yoon et al. 2020 [72]	Prospective longitudinal study	18	LI-MLD Asymptomatic (*n* = 11) Symptomatic (*n* = 7)	NR	Asymptomatic patients had significantly better cognitive and language skills than symptomatic patients who experienced regression similar to untreated patients

^a^ Publications by Fumagalli et al. [15,47], Zambon et al. 2025 [48] and Faccioli et al. 2023 [49] include overlapping patient populations (see text). The number of unique patients treated with arsa-cel are as follows: PS LI-MLD (*n* = 19); PS EJ-MLD (*n* = 8); ES EJ-MLD (*n* = 11). ^b^ One patient with LI-MLD was pre-symptomatic at time of enrolment but symptomatic at time of treatment. ^c^ Number of patients included in the integrated summary of efficacy (out of 39 treated patients, two were excluded from the integrated summary of efficacy). ^d^ Median age at disease onset (range), in mos: 64 (29–83).

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
