# Peer review of "Newborn Screening for Metachromatic Leukodystrophy: A Systematic Literature Review"

_2409-515X, 2025, doi:10.3390/ijns11040103_

Round 1
Reviewer 1 Report
Comments and Suggestions for Authors
This manuscript is a systematic literature review evaluating the current evidence for implementing newborn screening (NBS) for metachromatic leukodystrophy (MLD). It focuses on four key areas: performance of screening assays, diagnostic confirmation methods, feasibility of population-based screening, and impact of early diagnosis and treatment. A total of 52 studies were included, encompassing full-text articles and conference abstracts. The authors conclude that MLD meets the Wilson and Jungner criteria for NBS inclusion, citing reliable screening algorithms, subtype prediction, and improved outcomes through early treatment (gene therapy or HSCT).
MAJOR REVISIONS REQUIRED
Include a Quantitative Analysis of Sensitivity and Specificity:
The manuscript claims that screening for MLD has good sensitivity and specificity (e.g., in Figure 2), but does not provide aggregated quantitative data to support this. A meta-analysis or at least a summary table reporting sensitivity, specificity, PPV, and NPV across studies should be included, where data are available.
Exclude Conference Abstracts from the Systematic Review:
Abstracts (n = 12) were included despite the authors acknowledging that their methodological quality could not be assessed. As per PRISMA guidelines and standards of scientific rigor, studies without peer-reviewed full-text data should not be included in evidence synthesis.
Conflict of Interest Discussion:
One of the senior authors is affiliated with Orchard Therapeutics, a company that manufactures the gene therapy (arsa-cel) discussed extensively in the manuscript. The potential influence of this affiliation on study selection, interpretation of outcomes, and recommendations should be transparently discussed in the manuscript.
Condense the Results Section:
The results section is excessively long and often redundant, hindering readability. A more concise synthesis of key findings would improve clarity and highlight the most impactful data.
Remove Glossary-Like Paragraph (Lines 544–550):
This section appears to serve as a glossary of acronyms and definitions, which is inappropriate within the main text. Such content should be moved to other appropiate place.
Add a Detailed Limitations Section in the Discussion:
The discussion currently lacks a critical appraisal of the limitations of the review. Suggested additions include:
High heterogeneity across studies (e.g., different sulfatide biomarkers, cutoff values, and algorithms).
Risk of publication bias and overrepresentation of positive findings.
Inherent bias in included observational and retrospective studies.
Lack of randomized controlled trials.
Author Response
Dear Reviewers,
In addition to the changes outlined below, we have corrected minor typos and inconsistencies through the anuscript and endevored to improve the flow.
Reviewer 1
This manuscript is a systematic literature review evaluating the current evidence for implementing newborn screening (NBS) for metachromatic leukodystrophy (MLD). It focuses on four key areas: performance of screening assays, diagnostic confirmation methods, feasibility of population-based screening, and impact of early diagnosis and treatment. A total of 52 studies were included, encompassing full-text articles and conference abstracts. The authors conclude that MLD meets the Wilson and Jungner criteria for NBS inclusion, citing reliable screening algorithms, subtype prediction, and improved outcomes through early treatment (gene therapy or HSCT).
MAJOR REVISIONS REQUIRED
Include a Quantitative Analysis of Sensitivity and Specificity:
The manuscript claims that screening for MLD has good sensitivity and specificity (e.g., in Figure 2), but does not provide aggregated quantitative data to support this. A meta-analysis or at least a summary table reporting sensitivity, specificity, PPV, and NPV across studies should be included, where data are available.
Response: We agree that the text in Figure 2 was misleading, and it has been revised to “MLD screening assays that utilize DBS and a multi-tiered algorithm are available and have been tested in several studies.” At present, a formal meta-analysis of performance characteristics is not feasible due to heterogeneity of the data and the rapidly evolving field of NBS for MLD where these measures are not yet known. For example, both Negative Predictive Value and Positive Predictive Value are impossible to determine from the publications because of insufficient data and study duration.
Changes in manuscript: Figure 2.
Exclude Conference Abstracts from the Systematic Review:
Abstracts (n = 12) were included despite the authors acknowledging that their methodological quality could not be assessed. As per PRISMA guidelines and standards of scientific rigor, studies without peer-reviewed full-text data should not be included in evidence synthesis.
Response: We appreciate the reviewer’s emphasis on methodological rigor and the importance of peer-reviewed full-text data in systematic reviews. We agree that full-text articles generally allow for more robust quality assessment and data extraction. However, in the context of MLD and NBS (a rapidly evolving field), conference abstracts often represent the earliest dissemination of emerging data, that may not yet be published in full. Furthermore, while PRISMA guidelines prioritize transparency and reproducibility, they do not prohibit the inclusion of conference abstracts (more details on the 2020 update can be found here, https://www.bmj.com/content/372/bmj.n71). Rather, they recommend that limitations in methodological assessment be clearly acknowledged, which we have done.
To present the breadth and depth of emerging data in NBS, we have included all abstracts where there is not a corresponding manuscript.
Change in manuscript: Results.
Conflict of Interest Discussion:
One of the senior authors is affiliated with Orchard Therapeutics, a company that manufactures the gene therapy (arsa-cel) discussed extensively in the manuscript. The potential influence of this affiliation on study selection, interpretation of outcomes, and recommendations should be transparently discussed in the manuscript.
Response: We thank the reviewer for this important observation, and we fully agree that transparency is essential. We would like to communicate some more information on this point: 1) The first draft of this work was completed by Charlotte Chanson under the supervision of Assaf Givati, Senior Lecturer in Public Health Education and Programme Director for the Master of Public Health (MPH) at King's College London, hence her position as a senior author. 2) There was a systematic risk of bias assessment performed by an independent person, Sarah Hodgkinson (Oxford PharmaGenesis), as reported in the Methods and Results. 3) All 6 authors, including those not affiliated with Orchard Therapeutics, had equal input to the design and execution of the review. 4) Independent review was performed by Dr. Soumeya Bekri prior to submission.
We have updated the Methods to include more details about the independent review and added a new paragraph to the Discussion explicitly addressing potential conflicts of interest.
Change in manuscript: Methods, Section 2.4; Discussion, Section 4.6
Condense the Results Section:
The results section is excessively long and often redundant, hindering readability. A more concise synthesis of key findings would improve clarity and highlight the most impactful data.
Response: We agree and have edited the Results section with the aim of reducing redundancy and enhancing readability. We have endeavored to do this, whilst maintaining a sufficiently detailed narrative of the historical development and interplay between the different studies, such that the salient findings and nuances in interpretation from the original publication remain clear. In order to improve clarity, we have also rearranged some aspects of the Results and modified the flow to make it easier to understand.
Change in manuscript: Results, Sections 3.2–3.5 condensed.
Remove Glossary-Like Paragraph (Lines 544–550):
This section appears to serve as a glossary of acronyms and definitions, which is inappropriate within the main text. Such content should be moved to other appropriate place.
Response: We agree. The glossary has been removed from the main text and placed in the dedicated Abbreviations section. Therefore, we removed the abbreviations listed in the footnotes of Table 1 as well as Table 2, and all abbreviations can be found in the Abbreviations section.
Change in manuscript: Abbreviations, Footnotes of Tables 1 & 2.
Add a Detailed Limitations Section in the Discussion:
The discussion currently lacks a critical appraisal of the limitations of the review. Suggested additions include: High heterogeneity across studies (e.g., different sulfatide biomarkers, cutoff values, and algorithms). Risk of publication bias and overrepresentation of positive findings. Inherent bias in included observational and retrospective studies. Lack of randomized controlled trials.
Response: We thank the reviewer for this valuable suggestion. We fully agree that a more explicit appraisal of limitations, including those inherent in emerging science, strengthens the manuscript. Accordingly, we have added a sentence to the Results about the abstracts that were included being only those with no corresponding full-text publication, and also added a dedicated Limitations section that discusses study heterogeneity (e.g., screening assays, cutoffs, and algorithms), potential publication bias, the predominance of observational and retrospective designs, and the absence of randomized controlled trials.
In regard to the design of the studies of outcomes of treatment with arsa-cel and HSCT, an external natural history patient cohort is often the only ethical and robust comparator option for studying a novel treatment for a rare disease with a high mortality rate and no available treatment other than best supportive care, and this was the design used for the MLD treatment studies. Conducting a randomised controlled trial or a prospective natural history study for a fatal rare disorder like MLD may simply not be feasible (Adang et al., 2024), and external control arms based on natural history data are viewed as acceptable in guidance issued by the FDA and EMA (Shore et al., 2024; EMA, 2015; Khachatryan et al., 2023).
Change in manuscript: Results; Discussion, Section 4.6
References
Adang LA, Sevagamoorthy A, Sherbini O, Fraser JL, Bonkowsky JL, Gavazzi F, D'Aiello R, Modesti NB, Yu E, Mutua S, Kotes E, Shults J, Vincent A, Emrick LT, Keller S, Van Haren KP, Woidill S, Barcelos I, Pizzino A, Schmidt JL, Eichler F, Fatemi A, Vanderver A. Longitudinal natural history studies based on real-world data in rare diseases: Opportunity and a novel approach. Mol Genet Metab. 2024 May;142(1):108453. doi: 10.1016/j.ymgme.2024.108453. Epub 2024 Mar 18. PMID: 38522179; PMCID: PMC11131438.
Asbreuk, M., Schoenmakers, D. H., Adang, L. A., Beerepoot, S., Bergner, C., Bley, A., Boelens, J. J., Bugiani, M., Calbi, V., Garcia-Cazorla, A., Eklund, E. A., Fumagalli, F., Gronborg, S. W., Groeschel, S., Van Hasselt, P. M., Hollak, C. E. M., Jones, S. A., de Koning, T. J., van Kuilenburg, A. B. P., . . . Wolf, N. I. (2025). Metachromatic Leukodystrophy: New Therapy Advancements and Emerging Research Directions. Neurology, 105(2), e213817.
Beerepoot S, Schoenmakers DH, Fumagalli F, Groeschel S, Schöls L, Schiffmann R, Wong S, Boespflug-Tanguy O, Sevin C, Nadjar Y, Bley A, Mochel F, Horn MA, Baldoli C, Locatelli S, Hengel H, Laugwitz L, Hollak CEM, Gieselmann V, van der Knaap MS, Wolf NI. ARSA Variants Associated With Cognitive Decline and Long-Term Preservation of Motor Function in Metachromatic Leukodystrophy. J Inherit Metab Dis. 2025 Sep;48(5):e70072. doi: 10.1002/jimd.70072. PMID: 40751594; PMCID: PMC12317651.
EMA, 2015. Demonstrating significant benefit of orphan medicines. EMA, 2015. Available at:https://www.ema.europa.eu/en/news/demonstrating-significant-benefit-orphan-medicines. [Accessed 30 July 2025].
Fumagalli F, Zambon AA, Rancoita PMV, Baldoli C, Canale S, Spiga I, Medaglini S, Penati R, Facchini M, Ciotti F, Sarzana M, Lorioli L, Cesani M, Natali Sora MG, Del Carro U, Cugnata F, Antonioli G, Recupero S, Calbi V, Di Serio C, Aiuti A, Biffi A, Sessa M. Metachromatic leukodystrophy: A single-center longitudinal study of 45 patients. J Inherit Metab Dis. 2021 Sep;44(5):1151-1164. doi: 10.1002/jimd.12388. Epub 2021 May 4. PMID: 33855715.
Kehrer, C., Elgün, S., Raabe, C., Böhringer, J., Beck-Wödl, S., Bevot, A., Kaiser, N., Schöls, L., Krägeloh-Mann, I., & Groeschel, S. (2021). Association of age at onset and first symptoms with disease progression in patients with metachromatic leukodystrophy. Neurology, 96(2), e255–e266.
Khachatryan et al., 2023. J Pharmacokinet Pharmacodyn. https://doi.org/10.1007/s10928-023-09858-8.
Shore et al. [eds.], 2024. Regulatory Processes for Rare Disease Drugs in the United States and European Union: Flexibilities and Collaborative Opportunities. Available at: https://www.ncbi.nlm.nih.gov/books/NBK609376/. [Accessed 30 July 2025].
Reviewer 2 Report
Comments and Suggestions for Authors
I thank the authors for this very well structured comprehensive review. It synthesizes evidence on newborn screening for a rare autosomal recessive disorder, Metachromatic Leukodystrophy. They make a summary in both narrative form and tables on the NBS pilot studies in different countries and continents and include both prospective and retrospective data. They also take into consideration the clinical utility, accuracy and systemic considerations of NBS for this rare disease. They also review in detail the clinical trial data, including report on outcomes. This data is thus very relevant for making a case for the benefits of early and timely implementation of NBS for this condition which satisfies the Wilson and Jungner criteria. The literature review is pertinent with the new treatment available (arsa cel gene therapy) which is now approved in several countries as well as previously used HSCT (depending on phenotype). The availability of genotype/phenotype and enzyme activity correlations with clinical presentation as well as monitoring strategies and care pathways for this condition also makes implementation feasible.
1. The authors could consider using the more common subclassification of phenotypes into the three main subtypes to improve alignment with the prevailing literature (including genereviews).
Author Response
Dear Reviewers,
In addition to the changes outlined below, we have corrected minor typos and inconsistencies through the anuscript and endevored to improve the flow.
Reviewer 2
Thank the authors for this very well-structured comprehensive review. It synthesizes evidence on newborn screening for a rare autosomal recessive disorder, Metachromatic Leukodystrophy. They make a summary in both narrative form and tables on the NBS pilot studies in different countries and continents and include both prospective and retrospective data. They also take into consideration the clinical utility, accuracy and systemic considerations of NBS for this rare disease. They also review in detail the clinical trial data, including report on outcomes. This data is thus very relevant for making a case for the benefits of early and timely implementation of NBS for this condition which satisfies the Wilson and Jungner criteria. The literature review is pertinent with the new treatment available (arsa cel gene therapy) which is now approved in several countries as well as previously used HSCT (depending on phenotype). The availability of genotype/phenotype and enzyme activity correlations with clinical presentation as well as monitoring strategies and care pathways for this condition also makes implementation feasible.
The authors could consider using the more common subclassification of phenotypes into the three main subtypes to improve alignment with the prevailing literature (including genereviews).
Response: We thank the reviewer for this remark. While we recognize that the three-subtype classification of MLD (late infantile, juvenile, adult) is widely cited in the older literature, more recent literature strongly supports the existence of separate early juvenile (EJ) and late juvenile (LJ) subtypes (Asbreuk et al., 2025; Kehrer et al., 2021). Therefore, we have kept the four-subtype subclassification in the manuscript. The LI and EJ forms of MLD are more similar to each other in terms of disease course and treatment options based on recent literature (Asbreuk et al., 2025; Kehrer et al., 2021). Thesubdivision of MLD into 4 subtypes provides a clearer framework for discussing newborn screening and therapeutic pathways.
Change in manuscript: None.
References
Asbreuk, M., Schoenmakers, D. H., Adang, L. A., Beerepoot, S., Bergner, C., Bley, A., Boelens, J. J., Bugiani, M., Calbi, V., Garcia-Cazorla, A., Eklund, E. A., Fumagalli, F., Gronborg, S. W., Groeschel, S., Van Hasselt, P. M., Hollak, C. E. M., Jones, S. A., de Koning, T. J., van Kuilenburg, A. B. P., . . . Wolf, N. I. (2025). Metachromatic Leukodystrophy: New Therapy Advancements and Emerging Research Directions. Neurology, 105(2), e213817.
Kehrer, C., Elgün, S., Raabe, C., Böhringer, J., Beck-Wödl, S., Bevot, A., Kaiser, N., Schöls, L., Krägeloh-Mann, I., & Groeschel, S. (2021). Association of age at onset and first symptoms with disease progression in patients with metachromatic leukodystrophy. Neurology, 96(2), e255–e266.
Reviewer 3 Report
Comments and Suggestions for Authors
Please find review comments in the attachments

Author Response
Dear Reviewers,
In addition to the changes outlined below, we have corrected minor typos and inconsistencies through the anuscript and endevored to improve the flow.
Reviewer 3
Review of Newborn Screening for Metachromatic Leukodystrophy: A Systematic Literature Review
Dear Editor,
The authors have submitted a well-written manuscript containing important information on metachromatic leukodystrophy (MLD), its treatment, and newborn screening (NBS). However, there are several major concerns that should be addressed.
Overall Comments
The manuscript includes a mixture of study types—feasibility studies, assay development, and prospective NBS pilots. This diversity is important but should be more clearly categorized and interpreted accordingly.
Response: We agree. Prospective pilot studies are now separated from feasibility/method development studies in a revised Table 1. We have also added subheadings into Section 3.2 to specify more clearly Retrospective Studies (section 3.2.1) and Prospective Studies (section 3.2.2). Additionally, in the Results, we have significantly revised the limitations section (4.6) to highlight heterogeneity and limited long-term data.
Change in manuscript: Results, Section 3.2; Table 1; Discussion, Section 4.6.
Moreover, several authors of this review are also co-authors of the included publications and are affiliated with companies directly involved in MLD treatment. This raises a risk of conflict of interest, potentially introducing bias in how the evidence is interpreted and presented.
Response: We thank the reviewer for this important observation, and we fully agree that transparency is essential. We would like to communicate some more information on this point: 1) The first draft of this work was completed by Charlotte Chanson under the supervision of Assaf Givati, Senior Lecturer in Public Health Education and Programme Director for the Master of Public Health (MPH) at King's College London, hence her position as a senior author. 2) There was a systematic risk of bias assessment performed by an independent person, Sarah Hodgkinson (Oxford PharmaGenesis), as reported in the Methods and Results. 3) All 6 authors, including those not affiliated with Orchard Therapeutics, had equal input to the design and execution of the review. 4) Independent review was performed by Dr. Soumeya Bekri prior to submission.
We have updated the Methods to include more details about the independent review and added a new paragraph to the Discussion explicitly addressing potential conflicts of interest.
Change in manuscript: Methods, Section 2.4; Discussion, Section 4.6
The review does not adequately discuss these limitations, nor does it address the limited public data available on the feasibility and long-term benefit of NBS for MLD.
Response: We believe that the new Limitations section (4.6) thoroughly explains the current evidence base for MLD NBS as well as the current limitations. The Wilson and Jungner criteria for population-based screening do not include demonstration of long-term benefit, as NBS is an ongoing process. Once there is an effective therapy and a feasible screening method it is appropriate to initiate wider screening, along with appropriate safeguards such as regular evaluation of program performance and outcomes. We have added text to the manuscript in section 4.4 to explain this.
Change in manuscript: Discussion, Section 4.4 and 4.6.
Abstract
Line 26: The authors state that subtype prediction tools are available for screening, referring to research publications. I recommend rephrasing this statement to: “There is ongoing work to develop tools to predict disease severity” or similar, to avoid overstating current capabilities.
Response: We agree. We revised this statement to: 'There is ongoing work to develop tools to predict disease severity and subtype.' This has also been changed throughout the manuscript in other instances where it may have been considered overstating current capabilities.
Change in manuscript: Abstract, Discussion.
Introduction
Line 90: The manuscript states that certain U.S. states have added MLD to their NBS panels. Please clarify whether these states have initiated screening or have only approved its inclusion.
Response: Clarifications have been made. We have adapted the text to state that these three US states have approved the inclusion of MLD on their NBS panels.
Changes in manuscript: Introduction.
Table 1
This table combines five studies under the heading “MLD newborn screening pilot studies,” despite the studies assessing very different aspects (e.g., feasibility, assay performance). I recommend that only the two actual prospective screening pilot studies be retained in this table. The remaining studies should be categorized separately—e.g., under “Method Development and Feasibility Studies.”
Response: Thank you for the suggestion. We understand the rationale in making a clear distinction between the different study types. We have revised Table 1 to add an additional header row categorizing the two different study types as “Method Development & Feasibility Studies” and “Prospective Pilot Studies”. We have also changed the title of the Table to remove pilot, so now it is “An overview of MLD newborn screening studies”. We added details on tiering, labs, and second samples where available throughout the manuscript.
Change in manuscript: Revised Table 1.
For each included study, please clarify: Whether all tiers were performed on the initial screening card.
Response: This has been added where possible to the text.
Changes in manuscript: Results, Section 3.2; Table 1.
- If samples were processed in a single laboratory or transported to other facilities.
Response: This information is not stated in the original literature therefore it is not included in the SLR.
Changes in manuscript: None
Whether a second dried blood spot (DBS), new sample, was used.
Response: This has been updated in Table 1.
Change in manuscript: Revised Table 1.
- The total turnaround time for the screening algorithm.
- Response: Turnaround times are not reported in any of the literature to date. It was included among outcomes that we were looking for, but no studies included this information. To our knowledge workflow analyses have not yet been conducted in a routine public health lab process, however they are expected to form important future publications.
Changes in manuscript: Discussion.
- Method development and feasibility are important topics and could be summarized and compared in a separate section or table.
Response: Please see response above in regard to Table 1 and the revisions to add a header clearly noting the method development & feasibility studies as separate from the prospective pilot studies.
Change in manuscript: Revised Table 1.
- A row indicating which eligibility criteria each study contributes to would also enhance the clarity of the table.
- Response: For the NBS studies if it is not in the original study, we did not include this data.
Changes in manuscript: None.
Methods
Line 134: Please clarify whether studies were included if they reported only one or some of the eligibility criteria.
Response: We thank the reviewer for pointing out the need for clarification. We have revised the Methods section to make explicit that studies were eligible for inclusion if they reported at least one of the predefined eligibility criteria, even if not all criteria were addressed. This ensures transparency in the selection process while maintaining a comprehensive evidence base. Studies were considered eligible if they addressed at least one of the predefined eligibility criteria (e.g., screening assay performance, confirmatory diagnostics, feasibility, or clinical outcomes), even if not all criteria were reported.
Change in manuscript: Methods, Section 2.2.
Abbreviations
Lines 350–353 and 544–550: These appear to be abbreviations used in the text. Please move them to a dedicated “Abbreviations” section.
Response: We have moved all abbreviations from the footnotes of Table 1 and 2 to a separate Abbreviations section.
Change in manuscript: Table 1 footnotes; Table 2 footnotes; Abbreviations section.
Line 405 : The sentence begins: “Following early results from ScreenPlus…” Please provide: A brief summary of how many children have been screened in this project. A reference to support the reported results.
Response: We have updated this to include the most recent published data from the abstract of Clarke et al. presented at WORLD 2025: as of August 2024, 21,561 infants were enrolled in ScreenPlus.
Changes in manuscript: Results, Section 3.4.
Table 2: Please include information on quality of life if any of the included studies assessed this.
Response: Quality of life outcomes were not reported in the publications summarized here, so we have clarified this in the Results by adding a sentence: “None of the publications reported quality of life outcomes.”
Changes in manuscript: Results, Section 3.5.
Under “follow-up time,” please report minimum and maximum follow-up durations in addition to the median.
Response: This has been added to Table 2.
Changes in manuscript: Table 2.
Include age of disease onset (minimum, median, and maximum) for all studies reporting groups of patients. This is critical for assessing the duration of monitoring needed for children identified through NBS who are not eligible for arsa-cel.
Response: We have added the additional details, where available in the original publications.
Changes in manuscript: Table 2.
Add a row for study limitations, such as: Lack of cognitive outcome data. Absence of quality-of-life measures.
Response: A sentence has been added to Section 3.5 noting that none of the treatment study publications reported quality-of-life outcomes. We also added to the Discussion that it will be important for future studies to measure patient- and family-reported experiences, including psychosocial impact, healthcare use, and quality of life.
Changes in manuscript: Results, Section 3.5; Discussion, Section 4.7.
Missing control groups, if applicable.
Response: We have added to the Discussion: “Conducting a randomized controlled trial for a fatal rare disorder like MLD is neither feasible nor ethical, and the studies of outcomes of treatment with arsa-cel and HSCT utilized appropriate external natural history patient comparator groups, an approach that is accepted by health authorities.”
Changes in manuscript: Discussion, Section 4.6.
Since several of the included studies have overlapping patient populations, please state: The total number of unique treated patients.
Response: A footnote (d) has been added to Table 2 the total number of unique PSLI, PSEJ and ESEJ patients treated with arsa-cel
Changes in manuscript: Table 2, Footnotes.
A summary table showing total treatment outcomes and benefits, ideally broken down by MLD subtype, to accurately reflect efficacy.
Response: Table 2 summarizes the treatment outcomes by subtypes of MLD in detail.
Changes in manuscript: None.
Discussion
- The authors state that NBS for MLD “unequivocally satisfies the Wilson and Jungner criteria.” This assertion seems biased, especially in light of the extensive and potentially invasive monitoring required for children who are not eligible for arsa-cel, as described in the European consensus guidelines. I strongly suggest: Removing this statement from both the Discussion and the Abstract.
- Including a balanced discussion of the Wilson and Jungner criteria, including challenges related to monitoring, treatment eligibility, and ethical considerations.
Response: We revised the text to provide a balanced discussion, removing 'unequivocally' and adding challenges regarding monitoring, treatment eligibility, and ethics.
Change in manuscript: Discussion, Section 4.1.
- A section should be included discussing scenarios where the correlation between ARSA enzyme activity and disease severity is weaker in dried blood spot (DBS) testing compared to clinical diagnostics using whole blood. This discrepancy may lead to uncertainty in predicting disease severity, even when genetic analysis is used as a third-tier test, due to the presence of variants of uncertain significance (VUS) in the ARSA gene and the lower sensitivity of the enzymatic method used in the DBS context.
Response: The prediction of disease severity and subtype is done at the time of confirmatory diagnosis, based on a number of different assessments, not on ARSA activity in the dried blood spot. The level of ARSA activity in DBS is not expected to correlate with disease severity; this has been clarified in the text.
Change in manuscript: Results.
Conclusions
The statement that the evidence “strongly supports timely implementation of NBS for MLD” should be rephrased to reflect the limited amount of published data and the small number of cases identified through NBS so far.
Response: We revised to: 'Current evidence suggests feasibility and potential benefit, though cases identified through NBS remain limited and further data are needed.'
Change in manuscript: Conclusion.
References
Adang LA, Sevagamoorthy A, Sherbini O, Fraser JL, Bonkowsky JL, Gavazzi F, D'Aiello R, Modesti NB, Yu E, Mutua S, Kotes E, Shults J, Vincent A, Emrick LT, Keller S, Van Haren KP, Woidill S, Barcelos I, Pizzino A, Schmidt JL, Eichler F, Fatemi A, Vanderver A. Longitudinal natural history studies based on real-world data in rare diseases: Opportunity and a novel approach. Mol Genet Metab. 2024 May;142(1):108453. doi: 10.1016/j.ymgme.2024.108453. Epub 2024 Mar 18. PMID: 38522179; PMCID: PMC11131438.
Asbreuk, M., Schoenmakers, D. H., Adang, L. A., Beerepoot, S., Bergner, C., Bley, A., Boelens, J. J., Bugiani, M., Calbi, V., Garcia-Cazorla, A., Eklund, E. A., Fumagalli, F., Gronborg, S. W., Groeschel, S., Van Hasselt, P. M., Hollak, C. E. M., Jones, S. A., de Koning, T. J., van Kuilenburg, A. B. P., . . . Wolf, N. I. (2025). Metachromatic Leukodystrophy: New Therapy Advancements and Emerging Research Directions. Neurology, 105(2), e213817.
Beerepoot S, Schoenmakers DH, Fumagalli F, Groeschel S, Schöls L, Schiffmann R, Wong S, Boespflug-Tanguy O, Sevin C, Nadjar Y, Bley A, Mochel F, Horn MA, Baldoli C, Locatelli S, Hengel H, Laugwitz L, Hollak CEM, Gieselmann V, van der Knaap MS, Wolf NI. ARSA Variants Associated With Cognitive Decline and Long-Term Preservation of Motor Function in Metachromatic Leukodystrophy. J Inherit Metab Dis. 2025 Sep;48(5):e70072. doi: 10.1002/jimd.70072. PMID: 40751594; PMCID: PMC12317651.
EMA, 2015. Demonstrating significant benefit of orphan medicines. EMA, 2015. Available at:https://www.ema.europa.eu/en/news/demonstrating-significant-benefit-orphan-medicines. [Accessed 30 July 2025].
Fumagalli F, Zambon AA, Rancoita PMV, Baldoli C, Canale S, Spiga I, Medaglini S, Penati R, Facchini M, Ciotti F, Sarzana M, Lorioli L, Cesani M, Natali Sora MG, Del Carro U, Cugnata F, Antonioli G, Recupero S, Calbi V, Di Serio C, Aiuti A, Biffi A, Sessa M. Metachromatic leukodystrophy: A single-center longitudinal study of 45 patients. J Inherit Metab Dis. 2021 Sep;44(5):1151-1164. doi: 10.1002/jimd.12388. Epub 2021 May 4. PMID: 33855715.
Kehrer, C., Elgün, S., Raabe, C., Böhringer, J., Beck-Wödl, S., Bevot, A., Kaiser, N., Schöls, L., Krägeloh-Mann, I., & Groeschel, S. (2021). Association of age at onset and first symptoms with disease progression in patients with metachromatic leukodystrophy. Neurology, 96(2), e255–e266.
Khachatryan et al., 2023. J Pharmacokinet Pharmacodyn. https://doi.org/10.1007/s10928-023-09858-8.
Shore et al. [eds.], 2024. Regulatory Processes for Rare Disease Drugs in the United States and European Union: Flexibilities and Collaborative Opportunities. Available at: https://www.ncbi.nlm.nih.gov/books/NBK609376/. [Accessed 30 July 2025].
Round 2
Reviewer 3 Report
Comments and Suggestions for Authors
Nothing to add